# Development of the Third Generation of the Dual-Reciprocating Drill

**DOI:** 10.3390/biomimetics5030038

**Published:** 2020-08-06

**Authors:** Craig Pitcher, Mohamed Alkalla, Xavier Pang, Yang Gao

**Affiliations:** STAR LAB, Surrey Space Centre, University of Surrey, Guildford GU2 7XH, UK; m.alkalla@surrey.ac.uk (M.A.); k.pang@surrey.ac.uk (X.P.); yang.gao@surrey.ac.uk (Y.G.)

**Keywords:** dual-reciprocating drilling, subsurface sampling, integrated actuation mechanism, numerical simulation

## Abstract

The dual-reciprocating drill (DRD) is a low-mass alternative to traditional drilling techniques biologically inspired by the wood wasp ovipositor, which is used to drill into wood in order to lay its eggs. The DRD reciprocates two halves lined with backwards-facing teeth, enabling it to generate traction forces that reduce the required overhead penetration force. While previous research has focused on experimental testing of the drill’s operational and design parameters, numerical simulation techniques are being developed to allow the rapid testing of multiple designs, complementing and informing experimental testing campaigns. The latest DRD design iteration integrated a novel internal actuation mechanism and demonstrated the benefits of adding controlled lateral movements. This paper presents an exploration of how bit morphology affects drilling performance and a preliminary study of discrete element method (DEM) simulations for modelling DRD interactions in regolith. These have shown how regolith grain size and microscopic behaviour significantly affects the performance of different drill designs, and demonstrated how customisable drills can exploit the properties of various substrates. Two system prototypes are also being developed for the DRD’s third generation, each utilising novel actuation and sampling mechanisms. A final drill design will then be deployed from a planetary rover and perform the first DRD drilling and sampling operation.

## 1. Introduction

Drilling systems often play a critical role in planetary exploration missions. By accessing the subsurface of extraterrestrial bodies, they are able to make in-situ measurements, confirm observations made by remote sensing instruments, search for biomarkers indicating the signs of life and the presence of water-ice, acquire samples to be taken back to Earth, and more. While rotary drills are commonly used in terrestrial applications, large masses are needed to provide the overhead force (OHF) necessary to push the drill into the substrate. This makes them unsuitable for planetary exploration, given the stringent mass constraints imposed upon a mission and low gravity on other bodies such as the Moon and Mars [1]. Terrestrial percussive drilling involves vibrating the drill bit and is well-suited for high-strength and brittle rocks. Although it requires a lower overhead mass to operate, it has a poor penetration rate and is only effective at shallow depths [2]. Rotary-percussive drills are able to use the rotary technique to drill to significant depths, while the addition of percussive shocks reduces the overhead requirement. Despite being heavy, complex systems, they have been used on several missions, including ExoMars [3] and Curiosity [4]. Another percussive technique is the mole, which is a compact, self-propelling probe. Though only capable of drilling through regolith, it has been used on both the Beagle 2 [5] and InSight [6] missions.

Biomimetics, the implementation of methods used in nature to inspire novel solutions to engineering problems, has been used to create alternatives to established techniques in a wide range of fields, with applications in space ranging from swarm satellites to robotics [7,8]. This has also been applied to drilling systems, with the proposal of the Dual-Reciprocating Drill (DRD), a novel concept based upon the ovipositor of the *Sirex noctilio*, or wood wasp, which it uses to drill into wood in order to lay its eggs. The ovipositor consists of two halves, each lined with backwards-facing teeth, as shown in Figure 1 [9], which are reciprocated in opposing motions by the wasp’s abdomen muscles. As one half pulls upwards, the teeth engage with the surrounding substrate, creating a traction force that is then transferred to the descending half via the abdomen muscles to act as a penetration force, pulling it further into the substrate, thus reducing the need for masses to generate the OHF required for conventional drills. Additionally the reciprocating action, particularly the tension in one half, helps to stabilise the drill length against buckling, which allows the drill halves to be lighter and thinner, while the drill length itself can be made lighter as it is transmitting a compressive force equal to the tensile force generated by the stabilisation of the drill tip within the substrate. As such, the DRD presents a compact and lightweight drilling solution. The ovipositor has also inspired a number of drilling mechanisms in fields such as soft tissue surgery [10,11,12] and geology [13].

The work presented in this paper details the development of the third generation of the DRD system. This expands upon the work performed in the design and testing of previous iterations, by detailing the experimental analysis of the drill’s morphological design and introducing numerical simulation techniques being developed for the rapid modelling of multiple variations of the DRD bit. The drill has also reached the stage of development where consideration is being made for the instrument’s integration onto a planetary rover, while the addition of sampling mechanisms will further increase the value of the DRD as an alternative drilling technique.

### 1.1. Evolution of the DRD

The DRD was first conceived as part of a biologically-inspired planetary micro-penetrator concept. A simplified prototype was built and tested in chalk, mortar and clay, as shown in Figure 2. As well as demonstrating its capability for drilling into low-strength rocks, its drilling efficiency was shown to be comparable to other percussive drills [14,15].

#### 1.1.1. Parameters Defining the DRD Operation

The first generation of the DRD used an external test rig to convert the continuous rotary motion of a motor to a controlled reciprocating translation of two drill bit halves, as shown in Figure 3. This test bench set-up enabled the control and testing of the OHF acting upon the drill halves, as well as the reciprocation frequency, *f*, and amplitude, Δ. The Surrey Space Centre Mars Simulants SSC-1 and SSC-2 were used, with their densities determined by their preparation method, which involved either simply pouring the regolith from a fixed height, or by also including a vibrating base to compact the soil further. Additional details of the regolith properties and preparation methods are detailed in [16]. A series of tests revealed that increasing the *f* and OHF resulted in greater drilling depths, and that regolith preparation had a significant impact on performance [17].

Another series of experiments focused on the variables defining the shape of the drill bit itself. Sixteen plastic bit designs were used, allowing the full exploration of five geometrical parameters to depths of up to 760mm. Here, it was found that the major factors impacting the achievable drilling depth and speed were the total width of the drill and the angle of the cone. A degree of drill stem bending was also seen, with some of the bits even snapping at the stem connection point [19].

#### 1.1.2. Integrated Actuation Mechanism

A key observation of these experiments was the presence of slippage, in which the receding half, instead of gripping the regolith and remaining in place, would slide upwards by a distance of up to 90% that of the reciprocation amplitude [17]. Another notable observation was the presence of small lateral movements seen in the drilling motion, which were determined to be the cause of the additional depth gain the DRD achieves compared to static penetration [20]. These sideways movements anchor the receding half into the regolith, creating lateral forces at least an order of magnitude higher than the generated traction force.

To confirm the effects of lateral movements with regards to reducing slippage and improving drilling performance, a new actuation mechanism, known as the Dual Complex Motion Mechanism (DCMM), was designed. This had the goal of incorporating controllable lateral movements into the typical vertical reciprocation, creating a series of complex motions that can improve engagement with the regolith [21]. This is an evolution of the Quadruple Cam internal actuation mechanism design proposed in [22] and shown in Figure 4a, in which the rotary motion of a motor is converted by a cam wheel system to linear vertical reciprocation of drive rails attached to the drill heads. The DCMM expands upon this method by simultaneously laterally reciprocating an additional pair of drive rails, shown in Figure 4b,c. Depending on its set-up, the DCMM is able to create circular or diagonal motions, as well as typical vertical reciprocation.

The DCMM, as the second generation of the DRD, advances the design from the original proof-of-concept external test bench towards a system prototype that could potentially be deployed from a planetary rover. The concept follows the Quadruple Cam system architecture, in which the entire actuation mechanism is encased within the drill heads [22]. Experiments varying the reciprocation amplitudes of the DCMM, with the set-up shown in Figure 4d, confirmed that increasing the lateral reciprocation amplitude increases the drilling depth, and additional experiments performed with the test rig tilted at an angle also reached greater depths compared to drilling vertically [21]. The DCMM is best described as a research prototype, as although the design is a significant step towards a genuine drilling system, its focus on examining a wide range of drilling motions resulted in a design much larger in diameter and more complex than would ideally be used for a true system prototype.

## 2. Methods

Although the DRD continues to show potential as a planetary drilling solution, there is still significant work that must be performed before it becomes a viable alternative to current techniques. As the DRD design advances to the third generation, its continued development can be taken in two complimentary directions. The first is to further research into the parameters defining the DRD design and operation. While experimental analysis has significant value, and is continued here with the study of the drill bit morphological design, it is practically infeasible to experimentally test all of the parameter combinations. Although an initial study into numerical simulation techniques available at the time deemed them unable to satisfactorily replicate the forces acting on the drill bit [20], work is now also being performed to create a powerful tool that can produce accurate simulations of any drilling scenario in regolith without the need for physical experimentation. The methods for developing both the morphological drill bit design experiments and progressing the numerical modelling techniques is detailed here, with the results of both being detailed in Section 3.

Secondly, a true system prototype of the DRD, which can be integrated onto a rover for practical demonstrations of subsurface exploration, will be developed. This will allow operational factors such as site and drill selection, deployment, retrieval and autonomy to be addressed, whilst still allowing for the design and inclusion of novel mechanisms. A discussion detailing the development of two prototype third generation designs is given in Section 4. All of the experiments, simulations and designs detailed here were produced by STAR LAB within Surrey Space Centre at the University of Surrey in Guildford, UK.

### 2.1. Drill Bit Morphological Design

The flexibility of the ovipositor stem of *M. n. nortoni*, another wasp species, is believed to have a significant role in its drilling capability, particularly with hard formation surfaces such as wood tissue [9]. In order for the DRD to have a similar capability, it is necessary to study the morphology of the ovipositor.

From Figure 1b, it can be seen that the ovipositor has a curved profile with spiral teeth along its axis. The ovipositor profile shape is believed to have a significant role in improving the drilling capability, as seen in the experiments detailed by [19]. A further evolution based on shape optimisation of the drill bit itself is proposed here, and five different morphological designs, shown in Figure 5, were manufactured [18]. The design features of each are detailed in Table 1, along with the original drill bit used in [19], known as Bit O.

It should be noted that Bit 3 has features directly based on the wood wasp, with its helical teeth being very similar to that of the ovipositor. An interlock system was also added to this bit to avoid any splitting caused by the lateral forces on the helical teeth, similar to the olistheter used by the ovipositor to keep its valves connected, as shown in Figure 6.

The performance of these new designs was examined with the test rig used in Section 1.1.1 and shown in Figure 3a. To test the bits’ endurance in a wide range of substrate properties, the regolith simulants used were the fine-grained (30–90 μm) SSC-2 and the coarse-grained (300–800 μm) ES-3 [24]. Using an OHF of 3 kg, an Δ of 3 mm and a *f* of 2.8 Hz, the test rig is initially positioned so that the drill bits are held just above the regolith surface, before being released. The position of the rig is recorded via contact with a linear potentiometer attached to the support structure, through which the drilling depth is obtained. The results of these tests are presented in Section 3.1.

### 2.2. Numerical Modelling of the DRD

Development of the DRD has so far largely relied on extensive experimental testing. However, a complete understanding of the regolith and bit interactions, as well as the drilling motions, is still lacking. There is a vast range of potential drilling motions and geometrical, operational and substrate parameters, with each requiring new bit designs, actuation mechanisms and/or test rig set-ups to explore them. It is also a significant challenge to replicate the extraterrestrial environment, such as the atmospheric conditions and gravity, on Earth, as well as preparing the regolith simulants in a repeatable, robust and safe way. As such, optimising and understanding the full potential of the DRD design and its novel techniques using the experimental approach would be extremely cost-intensive and time-consuming. To address this, numerical modelling is considered to be a crucial tool for filling in these gaps in knowledge.

#### 2.2.1. Overview of the Simulation Techniques Used for Drill/Soil Interactions

Amongst the various simulation methodologies available, the Discrete Element Method (DEM) has been recognised as a promising tool for dissecting the physics of the deformation behaviour of regolith and predicting the tool/regolith interactions. DEM has been used to examine the macroscopic behaviour of regolith resulting from individual particle interactions in the microscopic scale. The fundamental physics of regolith, such as shape-dependent behaviour [25] and shear band formation [26], as well as practical aspects including regolith excavations [27], sampling processes [28,29] and wheel–regolith interactions [30,31,32], have also been explored. Additionally, DEM has been used for regolith drilling studies, including creating a heat transfer model [33], analysing penetration resistance [34] and for the design of lunar coring bits [35].

The only DRD-regolith simulation using DEM to date examined the tool–regolith interactions during the drilling process that are otherwise very difficult to observe experimentally. This study also examined the forces experienced by the DRD, which provided further insight into the relationship between penetration, traction and lateral forces as discussed in Section 1.1.2 [20]. Although it was demonstrated that the DEM is capable of producing large-scale models of drill–regolith interactions using one million particles, it was unable to produce similar forces to those seen in the experimental studies. Possible reasons for this include the absence of non-spherical particles used in the simulations, and that one million particles, a limitation of the available computational resources at the time, was insufficient, with at least three million particles being necessary [36].

#### 2.2.2. Preliminary Numerical Study

A preliminary study will first be performed to identify and develop appropriate contact models that can capture the behaviour of Martian and Lunar regolith simulants, such as SSC-2 and ES-3, and investigate how these different models affect the macroscopic response of the drilling process. Since the particle sizes of these simulants tend to lie in the range of 100–1000 μm, it is computationally expensive to develop a multi-million particle model with these sizes to simulate the drilling process. Proper granular scaling strategies have to be introduced to strike a balance between numerical accuracy and a realistic simulation time-frame under limited computational resources. The results of the first numerical modelling study using the EDEM software are presented in Section 3.2. 

### 2.3. Development of System Prototypes

One of the major design goals for building a system prototype at this stage of development is to implement mechanisms with novel technologies, allowing further experimental testing alongside demonstrations of full subsurface DRD operations. Given that there are a wide range of mechanisms that can drive the DRD, two system prototypes following different design philosophies are being developed simultaneously, which are discussed in detail in Section 4.

One component that will be included in both designs is a sampling mechanism. While the inclusion of these can greatly increase the value of a drilling system, to date there has only been one passive sampling device considered for the DRD [15]. Cuttings collection methods will be utilised in these designs, as the unique motion of the DRD makes implementing typical coring systems and the hammering techniques used by moles impossible.

## 3. Results

### 3.1. Drill Bit Morphological Design

Using the test rig set-up described in Section 2.1, each of the new bit designs, as well as the original Bit O, were operated four times in SSC-2 and ES-3. The times taken to reach the target depth of 760 mm in both regoliths, with the averages of the four results, are presented in Figure 7 [18].

These results show that Bits 1 and 4–6 were able to reach the target depth faster than Bit O in SSC-2, while all bits outperformed Bit O significantly in ES-3. The percentage reduction ratios in time compared to Bit O is given in Table 2. These quicker times have many potential benefits, including less bit corrosion and lower energy consumption required for reaching greater depths.

The slower drilling times in SSC-2 and faster times in ES-3 for Bits 2 and 3 show that their convex and helical teeth features, which are suited for cuttings removal, have a better traction capability over their penetration, and as such are useful for engaging with harder soil formations. Conversely, while the toothless, concave Bits 5 and 6 are significantly quicker than most other drills in SSC-2, the small increase in reduction ratio from SSC-2 to ES-3 suggests a better suitability in finer regoliths. Additionally, the high reduction ratios can be attributed to the absence of the normal burden of carrying regolith on the teeth during drilling rather than pushing the cuttings out of the bore hole.

These results have highlighted the necessity for having customisable drill bits for different grain sizes, which will be beneficial for different locations on planetary bodies. It has also demonstrated that each feature in the bit designs has its own individual influence that can be compatible with certain types of regoliths. This is summarised in Table 3.

Figure 8 shows the penetration profiles of each bit over time, with the profiles selected being those with the drilling time closest to the calculated average time, and gives a good indication of the performance of the drill bits at different depths. There are three notable segments on these curves. The initial penetration depth is solely a result of the test rig being released, with the weight of the OHF dropping the drill into the regolith. During the shallow penetration up to a depth of 500 mm, there is an almost linear relationship between the penetration depth and time, and all profiles are quite close to each other. By contrast, the relationship at depths beyond 500 mm is no longer linear, and there are noticeable differences between each profile [18]. The slippage rate of each drill is reflected in the penetration profiles, with less slippage resulting in a faster drilling time. While significant slippage is still present, it can be seen that all of the new bit designs have faster penetration rates, and thus have lower slippage, than Bit O in both regolith simulants.

### 3.2. Numerical Modelling of the DRD with EDEM

A simplified 3D numerical model of Bit 1 from Section 2.1 and 300,000 spherical particles with a 1mm diameter was developed in EDEM software, a commercial DEM package, as shown in Figure 9. A velocity-controlled loading was imposed with values of ±50 mms^−1^, with downwards motion being positive, to represent the reciprocal action of the drilling process. Two contact models, Hertz-Mindlin and Hertz-Mindlin with Johnson-Kendall-Roberts (JKR) cohesion, were considered in this study. The former is a standard contact model in which only normal and tangential contacts were considered, while the latter introduces a cohesion force between the contacts.

An initial comparison of the forces in the axial direction obtained from the two contact models as the DRD drills into the regolith is shown in Figure 10. It can be seen that the cohesion model exhibits a higher resistance both when drilling into and pulling out from the bulk particles. This shows that even when the drill geometry and loading conditions are the same, the microscopic interactions between the particles and the drill will have a significant impact on the macroscopic responses and should therefore not be neglected.

A comparison of the models after three seconds, at the point where the drill bit is being pulled out, is presented in Figure 11. It can be seen that there are more particles adhered to the drill bit wall in the Hertz-Mindlin with JKR model compared to the cohesionless model. This provides an insight as to why the former exhibits a higher drilling resistance, as the adhered particles could potentially diminish the effectiveness of the drill bit geometry during the penetration process.

The Hertz-Mindlin models used in this study provide a good approximation of the drilling forces. It should be noted that these are highly simplified models designed to demonstrate how the microscopic behaviour between particles can significantly affect the final macroscopic responses. At this stage, the parameters used in these contact models and the simulated results will not be calibrated against experimental data. However, they will be used instead as a benchmark when modelling the DRD process in future studies. A comprehensive study into regolith characterisation, DEM parameters and DRD contact model calibration is currently being conducted, which will capture the mechanical behaviour of the regolith more accurately and reduce the simulation error. This will involve taking into account various other parameters including particle shape, size distribution, and particle–particle and particle–wall friction forces.

## 4. Discussion

### 4.1. Development of the Internally Actuated System Prototype

This design continues with the philosophy chosen in [22] and uses a system architecture in which the entire actuation mechanism is contained within the drill heads. This results in a compact, self-contained system that can be deployed from a rover without the need for complex, heavy drill string systems. The drill will aim to have a size comparable to that of percussive moles, such as Insight’s HP^3^ [6] and Beagle 2’s PLUTO [5], which respectively have diameters of 27 mm and 20 mm, and lengths of 396 mm and 280 mm. The architecture chosen for this design will also be used to explore how different types of reciprocation may benefit the drilling performance.

The cam drive system previously used for the DCMM [21], shown in Figure 4, was determined to be too space inefficient. Instead a bi-directional screw, i.e., a lead screw with opposing threads to which sleeve nuts are attached, will be used. Lead screws are typically implemented in mechanisms that prevent the nuts from rotating. As such, when the screw is rotated by a motor, the nuts are forced to move linearly along the thread. By alternating the direction of the rotary motion of a DC motor, the rotation of the bi-directional screw will result in the nuts reciprocating linearly in opposing directions. The screw and direction of motion of the sleeve nuts is shown in Figure 12.

Each nut is connected to one of the drill heads via an interlocking mechanism. The position of this mechanism can be altered at specified points in the reciprocation cycle to decouple one of the halves, resulting in only one half reciprocating, known as single-half motion. The interlocking mechanism can then be moved further to have both drill heads attached to the same nut, creating a percussive motion where both halves reciprocate together in the same direction, effectively acting as a single drill head block. The possible motions achievable with this internal mechanism design are shown in Figure 13. The drill heads are held together using another passive interlocking system, with a circular protrusion in one half able to slide freely along a corresponding groove in the other half.

While typical percussive drills are effective for drilling into hard rock formations [37], the self-propelling mole designs are only capable of penetrating regolith. Unlike the moles, the DRD is assisted by an overhead force from the deployment mechanism, which may increase its effectiveness for drilling in harder materials, such as soft rocks and clays, compared to both moles and the original DRD design [14]. Using this percussive motion may therefore greatly increase the range of substrates the DRD is suited for drilling into.

When only one half is reciprocating, there is an imbalance of forces created, with the moving half being the only one producing a penetration force. The immobile half will only generate traction forces, and may act as a pivot for the drill to rotate around, thus changing the drill’s trajectory. While uncontrolled curved trajectories have been created before with the DRD [19], and steering needles with flexible tips based on the wood wasp have been studied [11,38], this would mark the first instance of controlled trajectory alterations with the DRD. Changes in direction could potentially range from small adjustments to correct errors that would otherwise require the drilling operation to be redone, to creating curved trajectories that enable the drill to reach targets otherwise unattainable with fixed paths, such as underneath boulders.

The sampling mechanism to be used in this design involves two shutters, one per drill head, that cover the entrances to a pair of compartments in each drill head cone. By rotating the shutters, each compartment can be opened independently, allowing up to four samples to be taken at any time during a single drilling operation. A cut-out of the drill head cone with the shutter is shown in Figure 14. This would enable studies to be made that can accurately show how the properties of the regolith at a specific location change with depth.

### 4.2. Development of the Dual Reciprocation Oscillation Drill (DROD) System Prototype

In the experiments performed in Section 3.1, there was one instance where the mechanism was very loose, causing unintentional vibration. This led to a higher penetration rate compared to the typical experiments. The drilling depth profiles for both experiments (i.e., with and without vibration), plotted in Figure 15, show a significant improvement in drilling time due to this vibration, and was noted as an area for future work [18].

This complements the results found in [21], further emphasising the benefits of lateral motions for improving the drilling performance. The design of this drill will take into consideration these observations to further reduce the effects of slippage. As such, the drilling system proposed here combines both reciprocation and vibration motions into a compact, simple design that can fully engage with the surrounding regolith.

The vibration mechanism is inspired by a fish’s caudal fin. Most fish retract and relax their body muscles on both sides for progressing forward, and the caudal fin acts as a final propelling element [39,40,41]. This fin mechanism is mimicked to produce an oscillation motion of the drill bits.

The reciprocation motion of this system is based on a cylindrical cam with dual followers. As the cam is continuously rotated by a DC motor, both followers move forwards and backwards, as shown in Figure 16, with an amplitude that is a function of the slope of the cam. The oscillation motion is created by utilising a double-faced wedge. As the reciprocating drill halves slide on the inclined wedge faces, they oscillate around hinged joints. With the aid of torsional springs, the oscillating parts maintain continuous and full contact with the wedge, as shown in Figure 17.

While the reciprocation mainly depends on the cam slope, the oscillation motion depends on both the cam slope and wedge angle. Figure 18 shows the drill bit in different positions in accordance with the continuous rotation of the cam, representing a complete drilling cycle. One of the benefits of this design is that the drilling motion can be easily transformed from reciprocation/oscillation mode to reciprocation only by replacing the wedge with a straight block and/or fixing the hinged joints.

This design will aim to counter the inward lateral motion caused by the soil reaction force against the bits, resulting in slippage, as reported in [17]. The action of the outward lateral motion generated by the oscillation will act against the inward motion of the soil, allowing a greater engagement with the surrounding regolith, thus reducing slippage and increasing the penetration rate and performance of the drill. This can be aided further by careful selection of the drill bit morphological design most suited for the regolith in the selected drilling location.

In contrast to the system described in Section 4.1, the actuation system shown in the overall DROD design in Figure 19 is located away from the soil. This is designed to protect the system by avoiding any interaction between the electric motors and the fine regolith particles. The drill has a long hollow stem capable of reaching a 2 m depth which works as a case for preventing any contamination of the system, as well as preventing the two reciprocating halves inside from being stuck due to excessive friction caused by the regolith. The drill stem can be adapted to be rigid or flexible like the wood wasp’s ovipositor, and it can be easily extended to fit any depths. Flexibility in the stem will allow a degree of steerability to be added which could allow the drill to take advantage of fissures in the substrate, potentially increasing the penetration depth. In addition, the system contains a sampling chamber with a volume of 173.55 cm^3^ located behind the drill bits. This compartment can be opened or closed at a certain depth by a shutter, which is operated by a linear motor located at the main actuation system.

#### Kinematic Analysis of the DROD

A combination of different reciprocation and oscillation amplitudes can be obtained by changing the cam slope and the wedge angle. This relationship is investigated by performing a kinematic analysis of the DROD. The global *OXY* and local *oxy* frames are located at points A and C, respectively, as shown in Figure 20. Points A, B and C here represent the axis of the hinged joint, contact between the bit and the wedge, and the bit tip, respectively. The follower’s linear displacement in the *x*-direction, based on the angular displacement of the cam, is given by *d = a/2 sinθ*, where *a* is the cam amplitude. Given that the stem moves longitudinally by *d*, the oscillation amplitude in the lateral direction, *ε*, at point B is *ε = d tanψ*, where *ψ* is the angle of the wedge. From this, the oscillation displacement at the tip, *δ*, can be derived:(1)$$\delta =L\sin\varphi =\frac{L}{l_{1}}\varepsilon$$
where *φ* is the oscillation angle measured from the neutral position, *L* is the distance between A and C, and *l_1_* is the distance between A and B. The motion system’s inputs and outputs can be coupled in this form:(2)$$\left[\begin{array}{c} d\\ \delta \end{array}\right]=\frac{a}{2}\left[\begin{array}{c} 1\\ \frac{L}{l_{1}}\tan\psi \end{array}\right]\sin\theta$$

The DROD kinematic system is considered as a planar system in the *xy*-plane with 1-DOF (one actuator is used to get the targeted motion and define the position of any point on it). To calculate the mobility of the system, the following mobility law can be applied:(3)$$DOF=3\left(N-1\right)-2L_{p}-H_{p}$$
where *N* is the number of links (*N* = 6), as shown in Figure 17 and Figure 18, with parts 1 and 7 considered as one link since they are both fixed within the drill frame, *L_p_* is the number of lower pair joints, such as the hinged and sliding joints (*L*_p_ = 5), and *H_p_* is the number of higher pair joints, such as the follower/cam and pin/wedge contacts (*H*_p_ = 4). During the motion of the drill bit, points A, B and C change location to A’, B’, and C’, with point A only moving in the longitudinal direction. The position of the origin of frame *oxy* with respect to *OXY* can be derived by:(4)$$\vec{R_{AC^{’}}}=\vec{R_{AA^{’}}}+\vec{R_{A^{\prime }C^{\prime }}}$$

Here, the orientation is defined as a rotation around the *z*-axis by *φ* as given in Equation (1). Hence, the position of the tip frame *oxy* and its orientation with respect to *OXY* are given as:(5)$$\left[\begin{array}{c} x\\ y\\ \varphi \end{array}\right]=\left[\begin{array}{c} d+L\cos\varphi \\ \delta +\frac{e}{2}\\ \tan^{-1}\left(\frac{\varepsilon }{\sqrt{l_{1}^{2}-\varepsilon^{2}}}\right) \end{array}\right]=\left[\begin{array}{c} \frac{a}{2}\sin\theta +L\frac{\sqrt{l_{1}^{2}-{\left(\frac{a}{2}\tan\psi \sin\theta \right)}^{2}}}{l_{1}}\\ \frac{aL}{2l_{1}}\tan\psi \sin\theta +\frac{e}{2}\\ \tan^{-1}\left(\frac{\frac{a}{2}\tan\psi \sin\theta }{\sqrt{l_{1}^{2}-{\left(\frac{a}{2}\tan\psi \sin\theta \right)}^{2}}}\right) \end{array}\right]$$
where *φ* is the angle between the *x*-axes of the local and global frames and *e* is the gap between the two tips. By defining the parameters of the DROD design, the tip pose can then be determined. Two different cylindrical cams (with amplitudes *a*_1_ = 6 mm and *a*_2_ = 8 mm) and wedges (with angles ψ_1_ = 8° and ψ_2_ = 5.8°) are considered, so that the effects of changing these parameters on the final bit motion, and consequently defining the dominant parameter for improving the drilling performance, can be investigated.

By using these parameters, the trajectory of the drill bits with different cam amplitudes and wedge angles is plotted in Figure 21. The oscillation displacement is not symmetrical around the *y*-axis, due to the existence of the gap between both bits. However, the neutral position at *x* = 282.5 mm is at the midpoint of all these trajectories, meaning that the oscillation in both sides is symmetrical. The velocity of the drill bit can be derived by differentiating Equation (5) by time:
(6)$$\left[\begin{array}{c} \dot{x}\\ \dot{y}\\ \dot{\varphi } \end{array}\right]=\left[\begin{array}{c} \frac{a}{2}\cos\theta -\frac{La^{2}\tan^{2}\psi \sin\theta \cos\theta }{l_{1}\sqrt{l_{1}^{2}-{\left(\frac{a}{2}\tan\psi \sin\theta \right)}^{2}}}\\ \frac{aL}{2l_{1}}\tan\psi \cos\theta \\ \frac{a\tan\psi \cos\theta }{2\sqrt{l_{1}^{2}-{\left(\frac{a}{2}\tan\psi \sin\theta \right)}^{2}}} \end{array}\right]\left[\dot{\theta }\right]$$
where $\dot{\theta }$ is the angular velocity of the cam. The linear velocities of the drill bits are plotted in the space of linear displacement in the *x* and *y* directions, as shown in Figure 22. Here it can be seen that the *x*-direction velocity is almost unaffected by the wedge angle. On the other hand, the velocity in the *y*-direction is affected by changing both the cam amplitude and the wedge angle. Both plots show that the maximum velocity is reached when passing through the neutral position; as this is the midpoint of the bit tip path, this is expected. One of the advantages of this model design is that the acceleration and jerk of the drill bits remain as sinusoidal functions; therefore, no high peak or jerk causing system instability or vibrations could affect the performance of the mechanism.

## 5. Conclusions

This paper has discussed various avenues of development undertaken for advancing the dual-reciprocating drill towards its third generation design. To do this, experimental-based research has taken place to examine the performance of six drill bits with unique morphological designs in regolith simulants. Using the original external test rig set-up, experiments were performed to examine drilling speed in SSC-2 and ES-3 in comparison to a previous drill bit design. Here, it was seen that the improvement in performance, if any, of the new designs compared to Bit O varied depending on the substrate being drilled into. The toothless Bits 5 and 6 showed a high capability in the fine-grained SSC-2, while Bits 2 and 3 showed much greater improvement in ES-3. This demonstrated how the shape of the drill can have a significant effect on performance depending on the substrate being drilled into, and the need for careful selection of drill bit design depending on the properties of the target drilling area.

The need for accurate numerical modelling techniques to enable the simulation of the vast combinations of DRD operational and design parameters in conjunction with physical experiments has now been addressed. The preliminary study here demonstrated how the microscopic behaviour between particles affects the final outcome of the macroscopic responses. This provides a basis for the future development of an accurate numerical model, in which a comprehensive understanding of parameters including particle size distribution, particle shape and friction is required.

Finally, two system prototypes are currently in development, each with novel actuation mechanisms and sampling subsystems. The first design will enable the DRD to change its dual-reciprocating motion to either percussion or single-half reciprocation. This could potentially allow the DRD to both drill into a wider range of harder substrates and create curved trajectories to reach targets that were previously unobtainable. The DROD design aims to develop a passive mechanism that can fit within a reasonably-sized drill capable of producing controlled lateral oscillations using an internal wedge and hinged bit design, allowing the creation of complex motions. The drill stem also has a degree of steerability that will be in the scope of further development for the third generation DRD. Each design also contains the first sampling mechanism to be implemented into the DRD, which will vastly increase the value of the drill when being considered for future missions. These designs will be built and integrated onto a planetary rover, leading to the first testing of a DRD system prototype and a demonstration of the drilling capabilities of the actuation mechanisms.

## Figures and Tables

**Figure 1 biomimetics-05-00038-f001:**
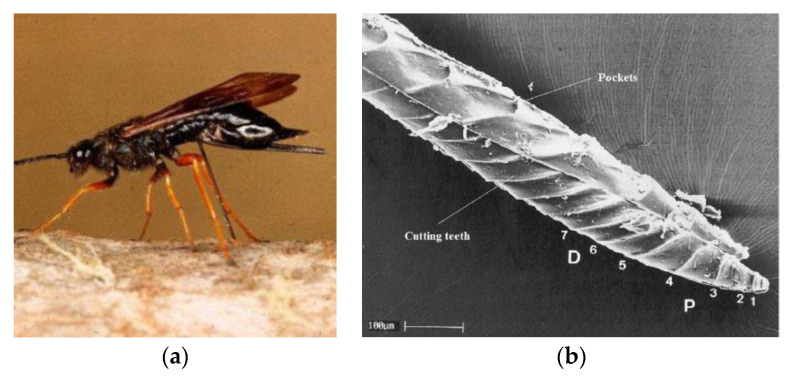
(**a**) The *Sirex noctilio* [14] and (**b**) a cross-section of its ovipositor, showing the proximally (**P**) and distally (**D**) pointing teeth **1**–**7** [9].

**Figure 2 biomimetics-05-00038-f002:**
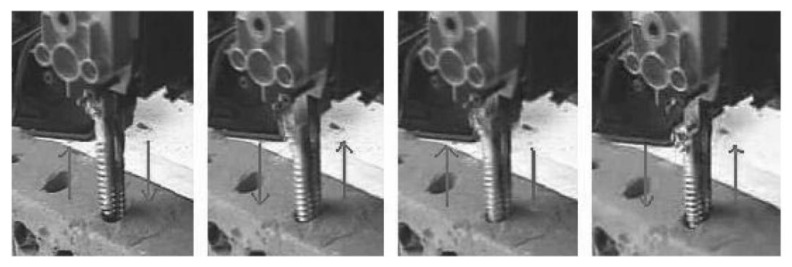
Lab-based test model [15].

**Figure 3 biomimetics-05-00038-f003:**
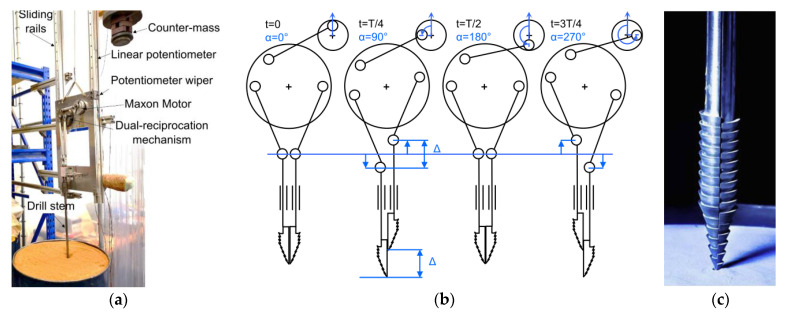
(**a**) Picture of the test bench set-up [18], (**b**) a schematic showing one full reciprocation cycle with the double pin and crank rocker mechanism, and (**c**) a picture of the DRD bit halves positioned above the regolith simulant [17].

**Figure 4 biomimetics-05-00038-f004:**
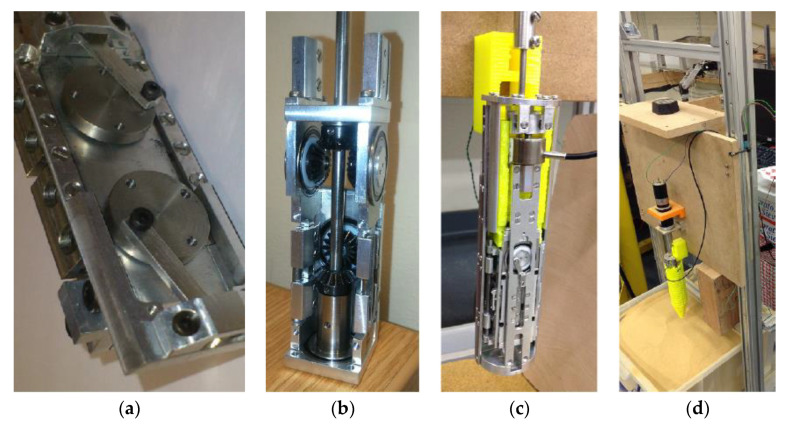
Pictures of (**a**) the original Quadruple Cam drive shaft [22], (**b**,**c**) the DCMM internal actuation mechanism [21] and (**d**) the test bench set-up.

**Figure 5 biomimetics-05-00038-f005:**
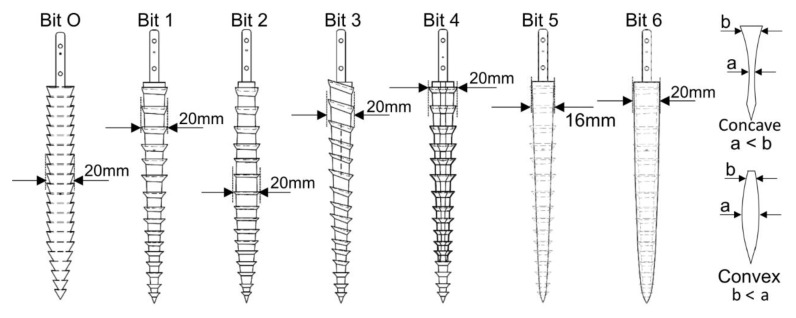
Proposed morphological designs for the drill bits [18].

**Figure 6 biomimetics-05-00038-f006:**
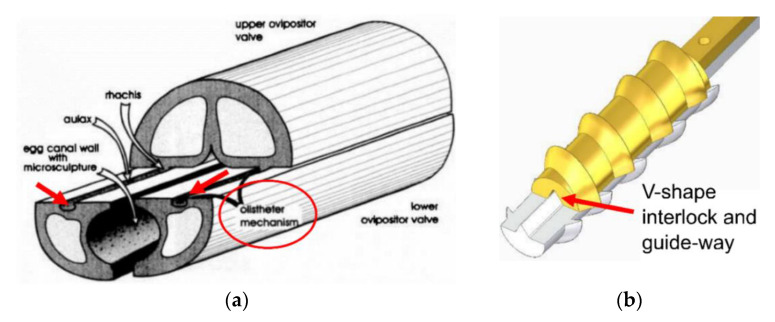
(**a**) Diagram of the ovipositor with the valves differentially protruding with the olistheter [23] and (**b**) the proposed V-shape interlock system for Bit 3 [18].

**Figure 7 biomimetics-05-00038-f007:**
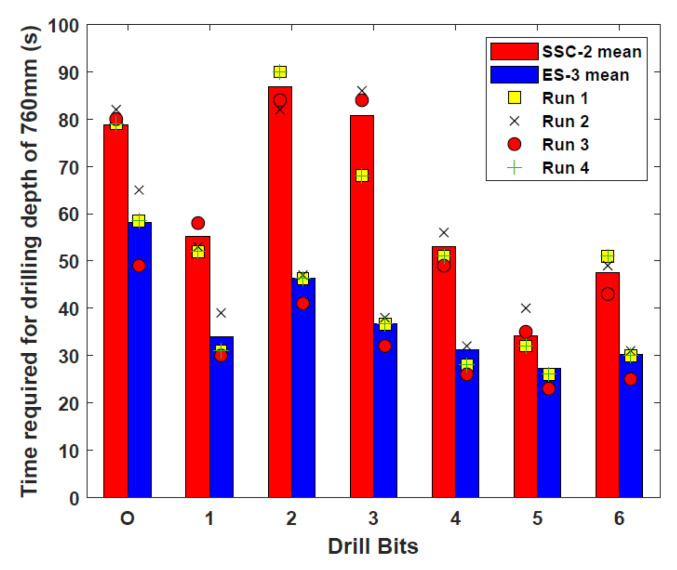
The drilling times and averages for all Bits in SSC-2 and ES-3. Adapted from [18].

**Figure 8 biomimetics-05-00038-f008:**
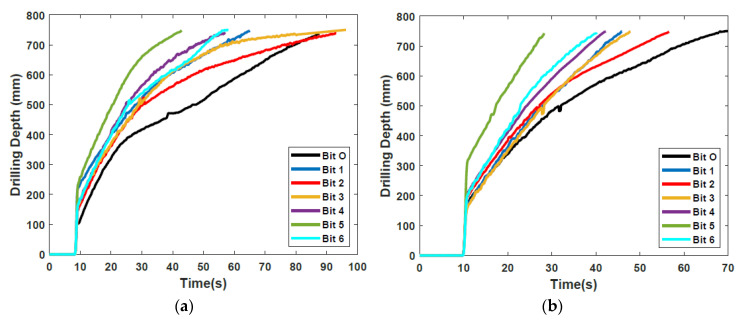
Drilling depth profile vs. time for all Bits in (**a**) SSC-2 and (**b**) ES-3. Adapted from [18].

**Figure 9 biomimetics-05-00038-f009:**
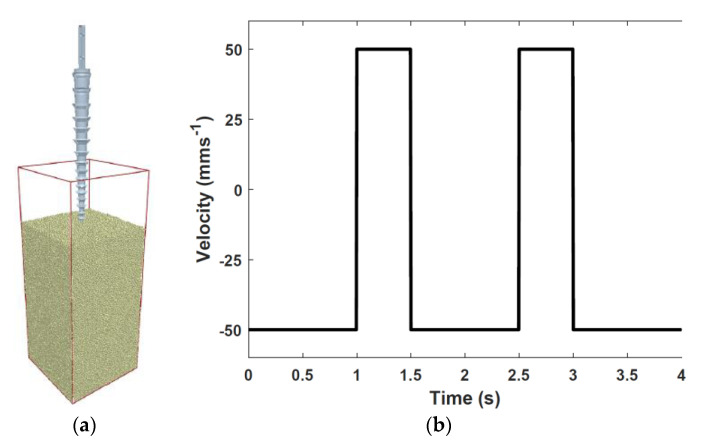
(**a**) Numerical model of Bit 1 and (**b**) the loading profile in terms of imposed velocity on the bit.

**Figure 10 biomimetics-05-00038-f010:**
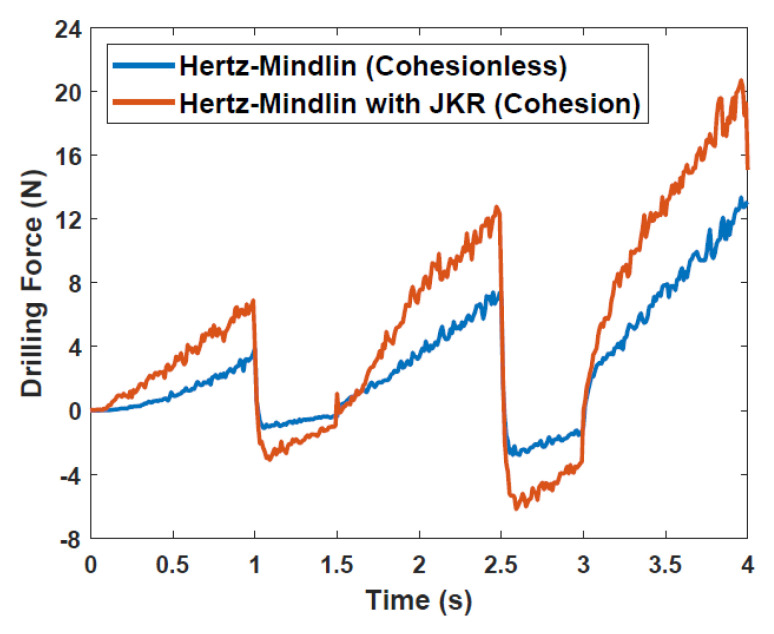
Drilling forces prediction with different contact models.

**Figure 11 biomimetics-05-00038-f011:**
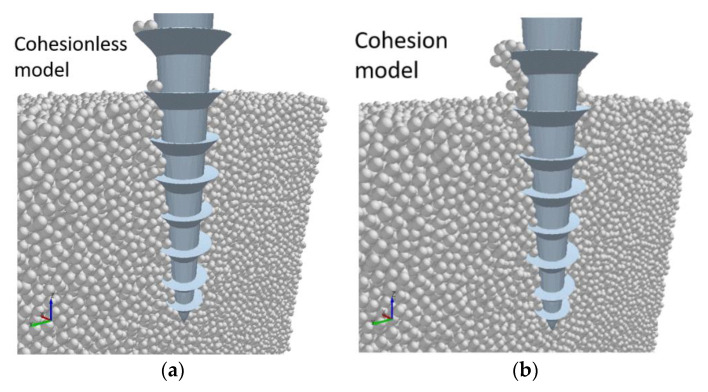
Comparison of the drill–regolith interactions at *t* = 3 s with (**a**) the Hertz-Mindlin and (**b**) Hertz-Mindlin with JKR contact models.

**Figure 12 biomimetics-05-00038-f012:**
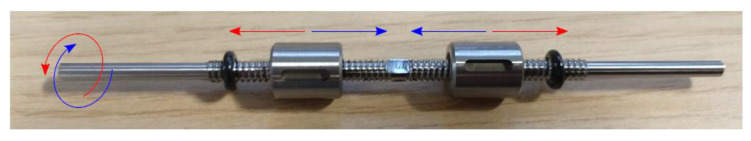
Picture of the bi-directional screw and sleeve nuts, with the red and blue translation arrows of the sleeve nuts corresponding to clockwise and anti-clockwise rotation of the screw respectively.

**Figure 13 biomimetics-05-00038-f013:**
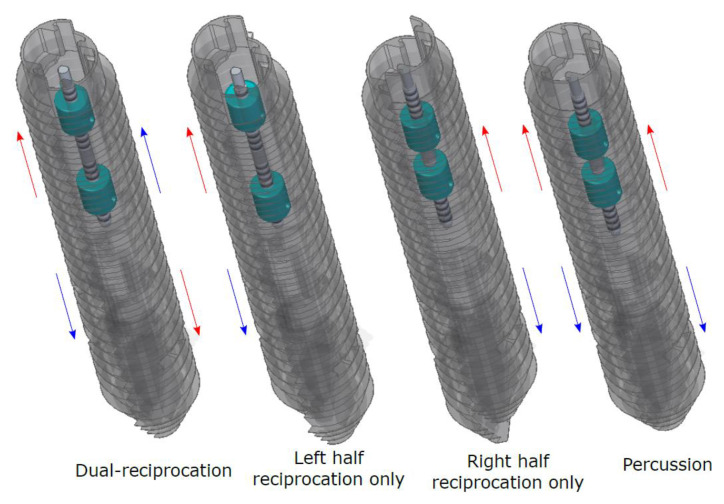
Drill head motions that can be created by the interlocking mechanism.

**Figure 14 biomimetics-05-00038-f014:**
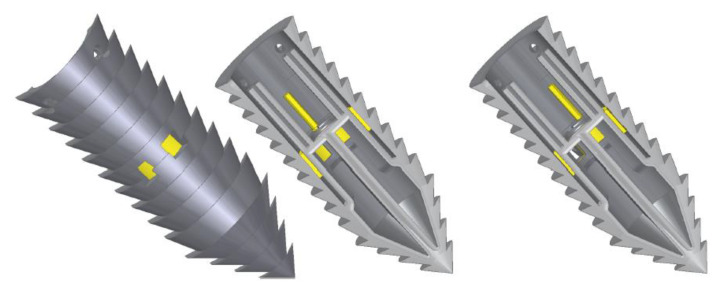
CAD cut-out model showing the sampling mechanism’s compartments in the drill head cone and the shutter in the closed and open positions.

**Figure 15 biomimetics-05-00038-f015:**
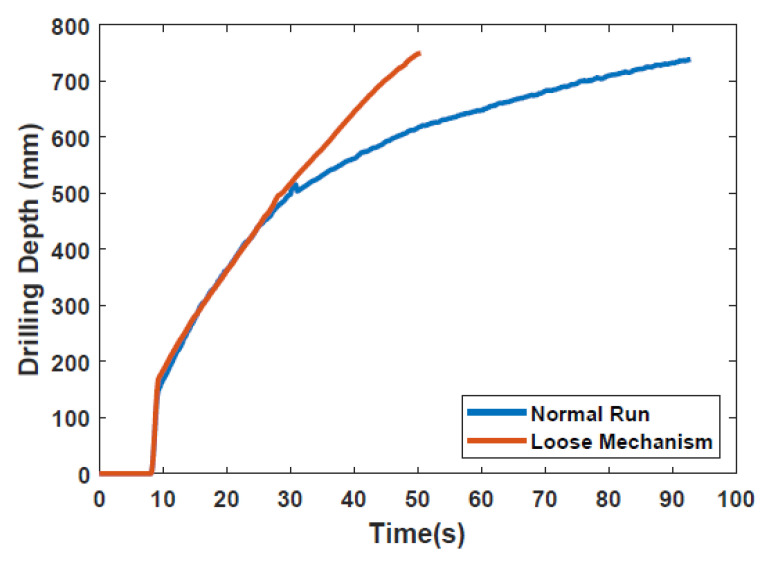
Comparison of Bit 3 runs in SSC-2 with a normal and loose mechanism.

**Figure 16 biomimetics-05-00038-f016:**
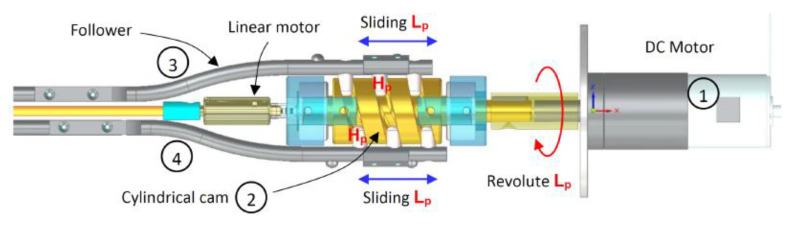
CAD model of the cylindrical cam with dual followers for the reciprocation motion, showing **1**: the DC motor as the fixed (ground) part, **2**: the cylindrical cam, and **3** and **4**: the followers.

**Figure 17 biomimetics-05-00038-f017:**
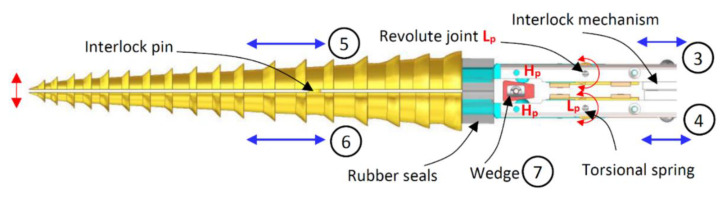
CAD model of the wedge at the end of the DROD for the oscillation motion, showing **3** and **4**: the followers, **5** and **6**: the oscillating bits, and **7**: the wedge as the fixed (ground) part.

**Figure 18 biomimetics-05-00038-f018:**
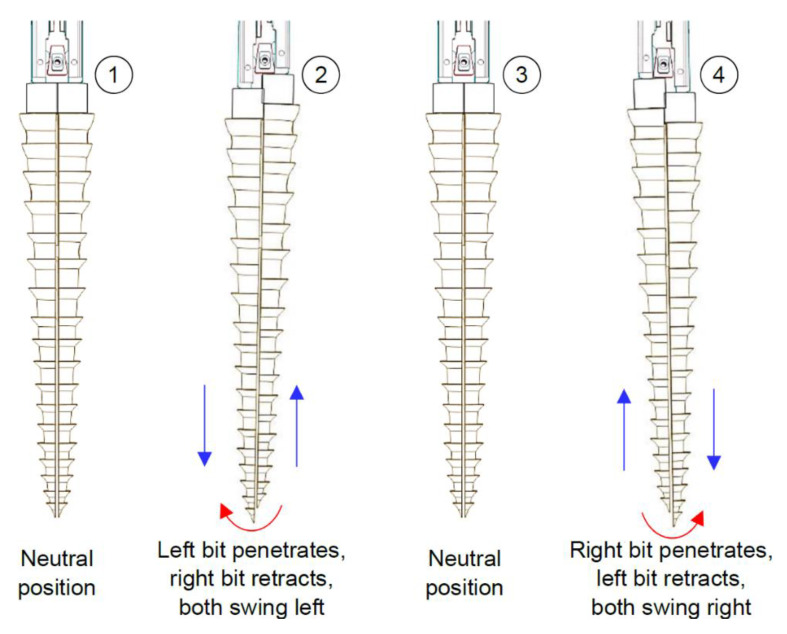
Synthesis of the reciprocation and oscillation processes, showing the movements of the drill as it performs one full reciprocation cycle.

**Figure 19 biomimetics-05-00038-f019:**
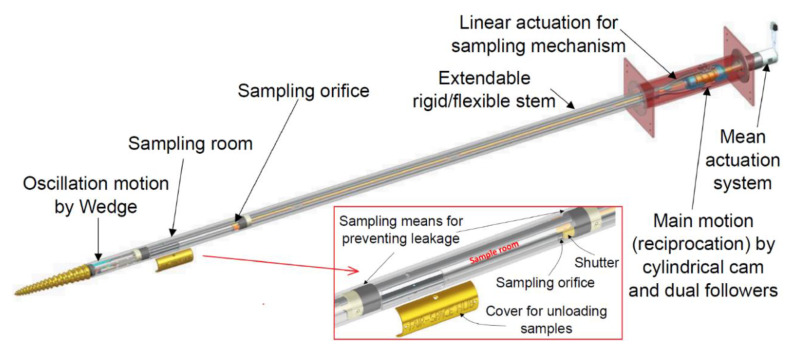
Overview of the DROD design.

**Figure 20 biomimetics-05-00038-f020:**
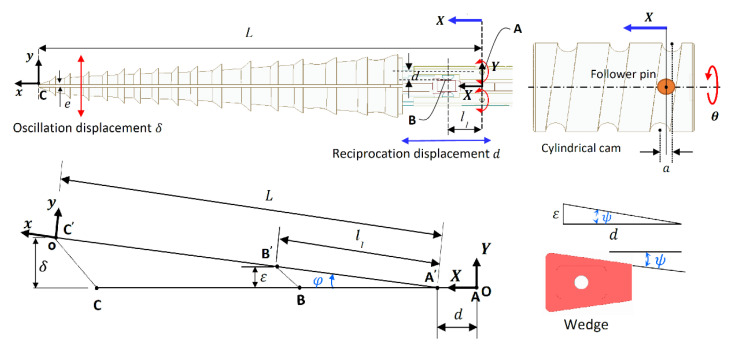
Geometrical analysis of the DROD.

**Figure 21 biomimetics-05-00038-f021:**
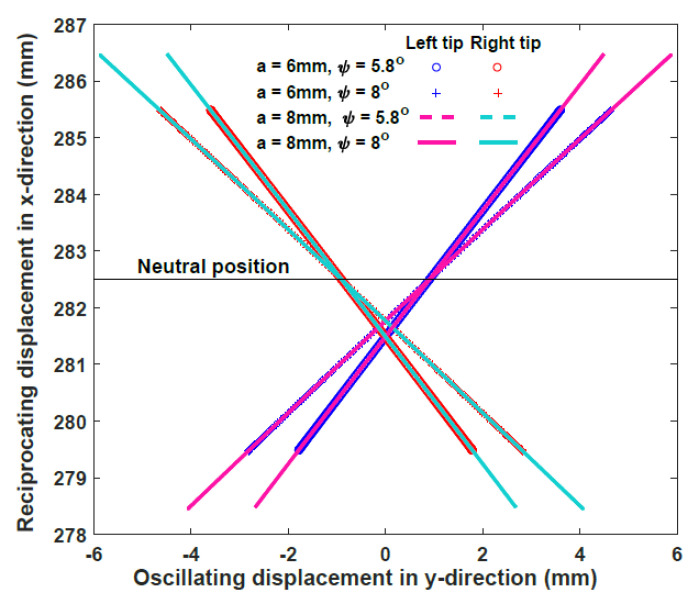
Trajectory of the left and right bit tips with respect to the global frame *OXY*.

**Figure 22 biomimetics-05-00038-f022:**
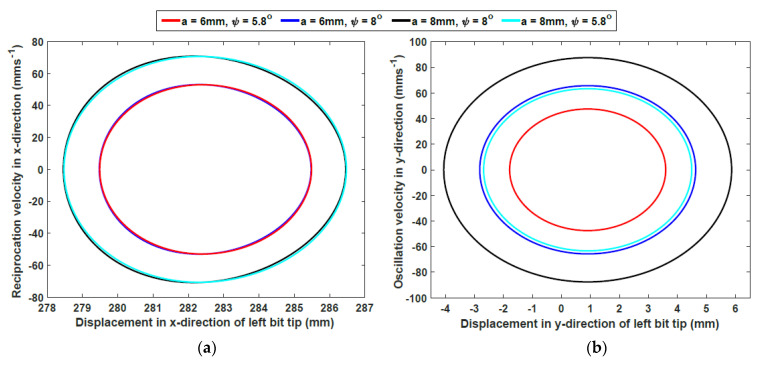
Plots of the bit tip’s (**a**) linear velocity $\dot{x}$ in the linear displacement space *x* and (**b**) linear velocity $\dot{y}$ in the linear displacement space *y*.

**Table 1 biomimetics-05-00038-t001:** Design features of the drills. Each bit has a height of 160 mm [18].

Drill Bit	Surface Profile	Teeth Angle	Cross-Section Contour	Half Bit Volume ×10^3^ (mm^3^)
O	Cylindrical, conical tip	Straight, H = 0°	Circular	15.75
1	Concave	Straight, H = 0°	Circular	9.65
2	Convex	Straight, H = 0°	Circular	11.63
3	Concave	Helical, H_max_ = 20°	Circular	9.32
4	Concave	Straight, H = 0°	Rhombic	7.891
5	Concave	No teeth	Circular	8.50
6	Concave	No teeth	Circular	15.26

**Table 2 biomimetics-05-00038-t002:** Reduction ratios of the drilling times for the Bits with respect to Bit O [18].

Drill Bit	SSC-2%	ES-3%
1	29.8	41.4
2	-	20.1
3	-	36.8
4	32.7	46.1
5	56.5	53
6	40.3	48

**Table 3 biomimetics-05-00038-t003:** Features of the proposed bits and their capabilities with different regoliths (+ + + Outstanding, + + Excellent, + Good, o Fair, – Poor, – – Bad) [18].

Design Features	Fine Grain Regolith	Coarse Grain Regolith	Potential Icy Regolith
Cylindrical profile	o	–	– –
Concave profile	+	+ +	+ +
Convex profile	– –	+	+ +
Circular cross-section	o	o	o
Diamond cross-section	+	+ +	+ + +
Helical teeth	– –	+ +	+ + +
Without teeth	+ + +	+	–

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
