# Peer review of "Development of the Third Generation of the Dual-Reciprocating Drill"

_biomimetics, 2020, doi:10.3390/biomimetics5030038_

Round 1

Reviewer 1 Report

This paper deals with the use of dual reciprocating drilling systems specially developed for planetary exploration missions. In particular the paper shows several studies and analysis that open the doors to the so called 3rd generation design of a DRD system.

This new generation is described in details and it is contextualized with reference to the main issues of the previous versions. Moreover, experimental-tests are presented for examining the performance of six drill bits with unique morphological designs. This part is a summary of what already shown in the paper:

  • Alkalla, M.G., Gao, Y., Bouton, A., “Customizable and optimized drill bits bio-inspired from wood-wasp ovipositor morphology for extraterrestrial surfaces”, (2019) IEEE/ASME International Conference on Advanced Intelligent Mechatronics, AIM, 2019-July, art. no. 8868816, pp. 430-435. DOI: 10.1109/AIM.2019.8868816

In any case the original part of this work is the one related to the design of the drilling solution that is presented in details but for which there is not an experimental phase to be analysed at the present stage of work. This is not a big lack but I wonder if the authors can address other considerations capable of sustaining the superiority of their approach in reducing the slippage phenomenon.

The paper is well written, structured, and clear. The literature review is complete even if the authors should extend their analysis also in other fields taking into consideration, for instance:

  • Nakajima, K., Schwarz, O. “How to use the ovipositor drilling mechanism of hymenoptera for developing a surgical instrument in biomimetic design”, (2014) International Journal of Design and Nature and Ecodynamics, 9 (3), pp. 177-189. DOI: 10.2495/DNE-V9-N3-177-189
  • Tarabini, M., Marzaroli, P., Saggin, B., Scaccabarozzi, D., Giberti, H., “Position uncertainty of a system for the localization of a reciprocating drill for geological inspections”, (2017) I2MTC 2017 - 2017 IEEE International Instrumentation and Measurement Technology Conference, Proceedings, art. no. 7969858, . DOI: 10.1109/I2MTC.2017.7969858

It is my considered opinion that, in the light of what is set out above, the paper is ready to be published with some minor revisions.

Author Response

Dear Sir/Madam,

     Following your review regarding the manuscript “Development of the Third Generation of the Dual-Reciprocating Drill” submitted to the Biomimetics journal for publication in a special issue, we are sending the rebuttal letter explaining the changes performed on the manuscript. We found your comments very useful, allowing us to elaborate on and clarify parts of the manuscript, and have made a number of modifications in accordance with the suggestions, which are detailed below.

  1. The reviewer noted that the section detailing the experiments that examined the performance of the six drill bits with unique morphological designs is a summary of what has been written in the paper “Customizable and optimized drill bits bio-inspired from wood-wasp ovipositor morphology” by (Alkalla et al, 2019).

The authors intended for Section 3 to indeed be a summary of the experiments described in this paper. This has been detailed as it is considered to be the first step of development in taking the DRD towards the third generation of the drill design. The paper had not been referenced correctly in the original draft, and this has been rectified, with references included in the text of Section 3, and in figures and tables where appropriate.

  1. The reviewer asked if it were possible to address any considerations that had been made in the new drill head designs that act to increase their performance with regards to reducing slippage.

The authors agreed that additional context should be given with regards to the designs’ attempts to reduce slippage, and this has been addressed throughout the paper. This is first done with additional descriptions in Lines 96 and 99-100 of how the initial addition of lateral movements combats slippage. The improved penetration rates of the new morphological designs over Bit O show a reduced level of slippage, which has now been detailed in Lines 187-190. The two system prototypes had different goals, with the purpose of the Single-Half and Percussive Motion system now clearly stated as being to explore the benefits of different types of reciprocation in Lines 287-289. As one of the major goals of the DROD system was to further improve the slippage reduction seen with the inclusion of lateral movements, this is now detailed in Lines 341-344 and a new paragraph in Lines 364-369.

  1. Finally, the reviewer has suggested that the literature review could be improved by exploring other fields that have used the ovipositor-inspired reciprocating motion.

The authors agree that the literature review could be further improved by this, and an additional sentence has been added in Lines 55-56 that references the use of ovipositor-inspired reciprocation in the fields of medicine and geology. This includes (Frasson et al, 2019), a paper that is also referenced in Line 318 in a discussion of flexible needle tip designs.

We sincerely appreciate the insightful and constructive comments and suggestions, and we believe that these have greatly strengthened the paper. Thank you again for taking the time to review this paper. Please contact me if you require any further clarifications.

Yours sincerely,

Craig Pitcher

Reviewer 2 Report

Recently, I reviewed the manuscript entitled “Development of the Third Generation of the Dual-Reciprocating Drill”. The content of this article is interesting and the authors described the work in detail. But I have some comments on this manuscript and suggest the authors to modify the content before it can be accepted for publication.

(1) This manuscript is very long. I suggest the authors to shorten the description of their previous work. So this work will focus on the new contribution and readers are easier to get the outline of this work.

(2) In line 77 on page 3: “Δ, and the preparation method of the drilled substrates”. Can you define more clearly about the “preparation method” you mentioned there? In line 187 on page 7, it has a unit of 3 mm. I am very strange why the “preparation method” can be measured by mm.

(3) Many pictures in this manuscript do not align centrally. The authors should avoid the space before the pictures, such as Figures 2 and 3 on page 3.

(4) Line 112 on page 4, the caption of Figure 4. There are four sub-figures, but the caption only defines three. Do the middle two denote the same DCMM?

(5) Line 172 on page 6. You defined seven morphological designs for the drill bits. So what are the heights of these bits? Did you develop all these bits? Can you present the developed prototypes here?

(6) How did you measure the depths in Figure 7? Can you give more details to show the experiments?

(7) In Figure 8, about in 10 seconds. Why the penetration depth increased suddenly from zero to more than 100 mm? How did this happen in your experiment? You mentioned in your manuscript that “The initial penetration depth, caused by releasing the test rig without consuming power or time, is different for each bit.” But I am still not able to get this process.

(8) In Figure 12, I think the right column is not necessary for the details are all included in the left column.

(9) In Figure 11, did you compare the penetration force in the simulation with that generated in actual penetration test? How about the error caused by simulation?

(10) The last comment is on the fonts of the equations, figures, and tables in this manuscript. The size varies greatly in different pages. I recommend the authors to modify them to the same size.

Author Response

Dear Sir/Madam,

     Following your review regarding the manuscript “Development of the Third Generation of the Dual-Reciprocating Drill” submitted to the Biomimetics journal for publication in a special issue, we are sending the rebuttal letter explaining the changes performed on the manuscript. We found your comments very useful, allowing us to elaborate on and clarify parts of the manuscript, and have made a number of modifications in accordance with the suggestions, which are detailed below.

  1. The reviewer noted that the manuscript is very long, and suggested shortening the description of the previous work to allow greater focus on the new contribution.

While the Biomimetics journal does not place any restrictions on length of the manuscript, the authors agreed that the previous work could indeed be shortened. As such, Sections 2 and 3 have been looked over and streamlined where possible, reducing the total length of the two sections by approximately one page.

  1. The reviewer asked that the preparation method be more clearly defined, and suggested that the preparation method was later given a unit of mm.

The authors have now added a brief description of the poured and vibration methods used in Lines 76-79, and direct the reader to the paper by (Gouache et al, 2010) for further details. The authors have also removed the confusion seen by the reviewer by modifying Line 75 to separate the Δ, used to represent the reciprocation amplitude, from the preparation methods.

  1. The reviewer noted that a number of figures did not align centrally, and that spaces should be avoided before the pictures.

The authors believe this was a result of the document being converted from the original Latex file to Word. The manuscript preparation guidelines for placing figures in Word has been consulted, and the figures now comply with this.

  1. The reviewer noted that only three captions had been used to describe four images in Figure 4, and asked for clarification.

The authors agree that this causes confusion, and have now added clear labels to this and all other figures that have multiple images individually referenced in the caption or main text.

  1. The reviewer asked for the heights to be given for the new morphological bit designs, and if the bits could be presented if they had been developed.

The authors have included the 160mm height of the bits, including Bit O, in the caption for Table 1 in Line 146. Although the authors do not have any photographs of the new bits, they were indeed manufactured, which has now been expressly stated in Line 143.

  1. The reviewer asked that extra details of the morphological experiments be given, including how the depth was recorded.

The authors have added further clarity to the experiments by replacing the schematic of the test bench set-up in Figure 3 (a) with a picture of the set-up used in these experiments, which includes the drill heads and measurement system, and is referred to in Line 154. An additional description as to how the depth was measured with a linear potentiometer has also been added in Lines 156-159.

  1. The reviewer requested clarity with regards to how the initial penetration depth increases suddenly from 0 to over 100mm.

The authors agreed that the description was not entirely clear. The initial increase is simply of a result of the test rig being released, dropping the drill into the regolith. An additional sentence has been added in Lines 183-184 which, coupled with the additions made to address Point 6, should now make this clearer.

  1. The reviewer suggested that the second column of Figure 8 was not necessary.

The authors agreed to remove the right column, and the two images left are now shown side-by-side.

  1. The reviewer asked if the penetration force in the simulation was compared with forces seen in the experimental tests, and wondered about the error in the simulation.

The authors have yet to test the parameters of the models against the experimental data at this point. However, a more detailed study is now underway which will more accurately reflect regolith behaviour and reduce simulation errors, which can then be used to make accurate comparisons. This has now been detailed in Lines 262-267.

  1. Finally, the reviewer requested that the size of the fonts of the equations, figures and tables be made more consistent throughout the manuscript.

The authors agreed that some changes could be made to make the fonts more consistent. The equations and tables have now been re-written in Word using the manuscript preparation guidelines. The text sizes for each graph in Matlab were the same, and is thus dependent on the size of the graph as it appears in the text. As the text is readable and remains consistent, the authors felt additional changes to the graphs are unnecessary. The size of the text in the other figures is also at the discretion of the size of the images as they appear in the manuscript, but the authors have made changes to make the font style consistent and easily readable in all images.

We sincerely appreciate the insightful and constructive comments and suggestions, and we believe that these have greatly strengthened the paper. Thank you again for taking the time to review this paper. Please contact me if you require any further clarifications.

Yours sincerely,

Craig Pitcher